# Utility of the Ratio between Lactate Dehydrogenase (LDH) Activity and Total Nucleated Cell Counts in Effusions (LDH/TNCC Ratio) for the Diagnosis of Feline Infectious Peritonitis (FIP)

**DOI:** 10.3390/ani12172262

**Published:** 2022-08-31

**Authors:** Pierpaolo Romanelli, Saverio Paltrinieri, Ugo Bonfanti, Marco Graziano Castaman, Elisa Monza, Walter Bertazzolo

**Affiliations:** 1MYLAV Veterinary Laboratory La Vallonea, 20017 Passirana di Rho, Italy; 2Department of Veterinary Medicine and Animal Sciences, University of Milan, 26900 Lodi, Italy

**Keywords:** cat, feline coronavirus (FCoV), clinical chemistry, effusion cytology, diagnostic accuracy

## Abstract

**Simple Summary:**

The concentration of lactate dehydrogenase (LDH) in the cavitary effusions from cats with feline infectious peritonitis (FIP) is high, as in other types of effusions, but the total nucleated cell count (TNCC) of the fluid is typically low in this disease. Therefore, we hypothesized that the LDH/TNCC ratio is higher in effusions from cats with FIP than in other effusions. To assess this hypothesis, the LDH/TNCC ratio recorded in 648 effusion of different types were classified based on the probability of FIP estimated through additional tests on fluids, blood or tissues. This approach confirmed that this ratio is higher in FIP effusions or with high probability of FIP. We provide some practical recommendation: when the laboratory receives and analyzes only the fluid, an LDH/TNCC ratio higher than 0.72 maximizes the possibility to correctly diagnose FIP (sensitivity and specificity of 79%). A ratio higher than 7.54 indicates a 6-fold higher probability to have FIP than another disease. Lower thresholds (0.54 and 2.27) have higher sensitivity and specificity (around 82%) respectively, or likelihood to have FIP (more than 10 times) when other changes consistent with FIP in blood or tissues are detected.

**Abstract:**

Background: We tested the hypothesis that the ratio between lactate dehydrogenase activity (LDH) and total nucleated cell counts (TNCC) in effusions may be useful to diagnose feline infectious peritonitis (FIP). Methods: LDH/TNCC ratio was retrospectively evaluated in 648 effusions grouped based on cytology and physicochemical analysis (step 1), on the probability of FIP estimated by additional tests on fluids (step 2) or on other biological samples (step 3, *n* = 471). Results of different steps were statistically compared. Receiver Operating Characteristic (ROC) curves were designed to assess whether the ratio identify the samples with FIP “probable/almost confirmed”. The cut-offs with the highest positive likelihood ratio (LR+) or Youden Index (YI) or with equal sensitivity and specificity were determined. Results: A high median LDH/TNCC ratio was found in FIP effusions (step1: 2.01) and with probable or almost confirmed FIP (step 2: 1.99; 2.20 respectively; step 3: 1.26; 2.30 respectively). The optimal cut-offs were 7.54 (LR+ 6.58), 0.62 (IY 0.67, sensitivity: 89.1%; specificity 77.7%), 0.72 (sensitivity and specificity: 79.2%) in step 2 and 2.27 (LR+ 10.39), 0.62 (IY 0.65, sensitivity: 82.1%; specificity 83.0%), 0.54 (sensitivity: 82.1%; specificity 81.9%) in step 3. Conclusions: a high LDH/TNCC ratio support a FIP diagnosis.

## 1. Introduction

Feline infectious peritonitis (FIP), the systemic disease induced by the feline coronavirus (FCoV), may have two main clinical presentations: dry (non-effusive) FIP is characterized by granulomatous lesions in one or more organs, whereas the wet (effusive) form is characterized by the presence of intracavitary effusions (abdominal or, less frequently, pleural or/and pericardial) [1,2]. While the diagnosis of dry FIP may be difficult—surgical biopsies followed by histology and immunohistochemistry for FCoV antigens are required to confirm the disease—the analysis of the effusions provide useful diagnostic information in the wet form: the fluids are usually yellowish, transparent to turbid and may contain fibrin clots. This gross appearance may be per se highly suggestive of FIP [3] but additional diagnostic information is provided by the physicochemical analysis of the effusion, including a high specific gravity (SG >1.015), a high protein content (>25 g/L), an albumin:globulin ratio <1.0 [4]. Protein electrophoresis of the effusion shows a similar pattern of serum protein electrophoresis, with an increased alpha2 and gamma-globulin, and a polyclonal profile [3]. The cell count is moderate to low (generally <5 × 10^9^/L) and the cytological examination of the fluid usually reveals a non-specific inflammatory pattern with a prevalent population of neutrophils (mostly non degenerated), in the absence of bacteria, and variable amounts of macrophages and lymphocytes. Another cytological finding highly consistent with FIP is the presence of proteinaceous granular background [5]. Although these findings increase the likelihood of FIP, other inflammatory conditions, characterized by protein-rich transudates or aseptic exudates, may mimic the physico-chemical and cytological profile of FIP. Among these inflammatory conditions, cholangitis-cholangiohepatitis and pancreatitis are the main differential diagnoses, along with any other condition causing intra-cavitary non septic inflammatory stimuli that may recruit neutrophils and increase the protein content of the effusions [1]. Recently, additional tests such as the Rivalta test [6,7], the shape of the scattergram and the delta-Total Nucleated Cells (TNC) measured with the laser-based automated cell counter Sysmex-iV [8,9] have been proposed as additional tests to support a clinical diagnosis of FIP. However, the Rivalta test is operator dependent [7], and the delta-TNC may be available only for users of the above-mentioned automated instrument. Moreover, both these tests may be falsely positive in cats that have effusions containing high molecular weight inflammatory proteins and may be falsely negative in FIP cats with atypically low protein concentrations in the effusions [7,8,10]. The likelihood of FIP may be very high when these findings are associated with other anamnestic, clinical and clinico-pathological findings consistent with FCoV infection such as: young age, FCoV-endemic environment, fever, jaundice, neurological signs, laboratory findings including normocytic normochromic anemia, microcytosis, neutrophilic leukocytosis with lymphopenia, hyperproteinemia with inverted A:G ratio, hypoalbuminemia and increased alpha 2 and gamma-globulin, increased concentration of acute phase proteins such as alpha1-acid glycoprotein or serum amyloid A [1,2,10,11]. In the absence of these additional clinico-pathologic findings, a conclusive diagnosis of FIP based only on conventional physicochemical or cytological findings of the effusion may be challenging. In commercial laboratories such information is often lacking and in most cases the effusion is the only sample available for the diagnosis. Therefore, an additional test on effusion samples that may further increase the probability of FIP when additional information or samples are not available, would allow to increase the pre-test probability and therefore address further confirmatory tests such as the polymerase chain reaction (PCR) for FCoV RNA. This latter test could be a valid confirmatory test that, regardless of the protocol used (e.g., qualitative, quantitative or associated with gene S sequencing), has been demonstrated to be very specific but poorly sensitive (50 to 70% of sensitivity in some studies) [10,12,13,14,15,16,17].

The activity of lactate dehydrogenase (LDH) in effusion samples has been considered since a long time as an useful biochemical test to discriminate canine neoplastic effusions from effusions with other pathogenic mechanisms [18] or, more recently, to discriminate exudates from transudates in cats [19,20] and in dogs [20,21], provided that samples are collected in vivo and not post-mortem [20]. In people, a high LDH activity in effusion or a high effusion/serum LDH ratio indicates a local production by inflammatory cells extravasated in the effusion [22,23]. LDH activity seems to not be useful in differentiating exudates with different pathogenesis (e.g., bacterial vs viral vs non septic). However, no studies investigated the possible role of the ratio between LDH and total nucleated cell counts (TNCC) in effusions. The hypothesis of our study is that the LDH/TNCC ratio may be useful to distinguish FIP effusions from other protein rich transudates or exudates, since the cellularity of effusions from FIP cats is usually low (similar to transudate) but the LDH activity is high (similar to exudates). The aims of this study were: (1) to retrospectively compare the LDH activity and the LDH/TNCC ratio recorded in different types of effusions collected from cats and submitted to a commercial diagnostic labs. To this aim, effusions have been classified either based on their cytological and physico-chemical pattern, or based on the likelihood of FIP formulated in light of additional information (signalment, hematology, clinical chemistry, histopathology, FCoV-PCR on effusion), when present. (2) to determine the diagnostic power of the LDH/TNCC in differentiating effusions with “probable” or “almost confirmed FIP” from other types of effusions. 

## 2. Materials and Methods

### 2.1. Study Design, Case Selection and Analytical Methods

This was a retrospective study based on the analysis of data available on the database of the laboratory Mylav. Specifically, the database was searched over a 5-year period, from April 2017 to March 2022. 

All the samples have been processed immediately upon arrival to the laboratory. More specifically, all the panel of tests requested by the submitting veterinarians were done on blood, if available, and effusions.

As regards effusions, at the time of analysis, the macroscopic appearance of the effusion was recorded and the specific gravity was measured using a refractometer (RETK-77, Tekcoplus Ltd., Hong Kong, China). The total nucleated cell concentration (TNCC) of the effusion, RBC and hematocrit were measured by an ADVIA 2120 (Siemens Healthineers, Dublin, Ireland). The following analytes were measured on the supernatant obtained by centrifugation at 1500 RPM (Eppendorf AG, Hamburg, Germany), using an automated spectrophotometer (Beckman AU5800, Tokyo, Japan) with dedicated biochemical kits: total bilirubin (3,5-dichlorophenyldiazonium tetrafluoroborate method), cholesterol (esterase/peroxidase method), triglycerides (glycerol phosphate oxidase method), urea (urease method), creatinine (Jaffe reaction), total protein (biuret method), albumin (bromocresol green method) and LDH (enzymatic-kinetic method). Cytological samples of the effusions were prepared by direct smear and by cyto-centrifugation at 1000 RPM for 5 minutes (Rotofix 32A, Hettich zentrifugen, Tuttlingen, Germany). Cytological slides were stained with May-Grunwald-Giemsa and were all examined by a board certified veterinary clinical pathologist (WB). Based on the results of the physico-chemical and cytological analyses, the most likely pathogenesis of each effusion was stated in the original report. The selection of cases to be included in this study was initially based on the final classification of effusions described in the reports. When more than one pathogenic mechanism was possible (e.g., hemorrhagic/neoplastic; lymphocyte-rich transudate/chylous, etc.), samples were excluded from the current study. Additional exclusion criteria were the lack of one or more information about cell counts or biochemical results.

The LDH/TNCC ratio was calculated on the basis of the LDH activity and the total nucleated cell counts included in the original reports.

The analysis of TNCC, LDH activity or LDH/TNCC ratio in the different types of samples was based on a 3-step approach.


*Step 1: Classification of samples based on physicochemical analysis and cytology of the effusions.*


Based on TNCC, chemical analyses and cytology, all the effusions were first classified as reported in Table 1, following the classification scheme proposed by Stockham and Scott [24] integrated with additional criteria proposed for feline effusions [25,26,27,28].


*Step 2: classification of samples based on the physicochemical and cytological probability of FIP on the basis of the effusion alone.*


The cytological descriptions were reviewed, along with chemical findings recorded on the effusions, signalment of the cats, cats age, and FCoV PCR, when available. This approach simulates what could happen in practice in diagnostic laboratories when the only tubes submitted for testing are the ones with anticoagulated effusion, and not blood or other tissues. The probability of FIP based on results of effusion was defined as reported in Table 2. 


*Step 3: classification of samples based on the overall probability of FIP on the basis of other biological specimens.*


As a further step of classification, the probability of FIP was reformulated on the basis of additional tests on blood or effusions, including serum protein electrophoresis, acute phase protein testing (serum amyloid A and haptoglobin), bacteriology of the fluid, FCoV PCR on fluids or tissues other than effusions (if not included in the original report), or histology/necropsies, if available. For this purpose, the database of the laboratory was screened to select those cases on which additional information potentially useful to classify the disease was available. If not, cases were excluded from this further step of analysis. Similarly, samples were excluded from the analysis when additional tests were not conclusive, i.e., when only non-specific findings (e.g., anemia, lymphopenia, biochemical changes consistent with organ damage or dysfunction) were available, in the absence of additional results highly suggestive of FIP (see below), when bacteriology was negative in samples classified as septic, or when FCoV PCR was negative, because this latter test is considered very specific but not sensitive for the diagnosis of FIP [10,12,13,14,15,16,17].

When additional information were present, cases were included in the analysis and the likelihood of FIP was reformulated based on results of:electrophoretogram: when normal, tended to exclude the diagnosis of FIP or allowed to classify samples as unlikely; conversely it supported the classification of samples as “possible” or “probable” based on the presence and magnitude of polyclonal gammopathy and/or of increased alpha2 globulins, especially if associated with other biochemical or hematological changes potentially consistent with FIP [10,29];acute phase proteins: normal concentration of serum amyloid A and/or haptoglobin were considered not consistent with FIP; increased concentrations, especially when associated with other biochemical, hematological or electrophoretic changes consistent with FIP, increased the probability of FIP up to “probable”, depending on the magnitude of the changes [30,31];bacteriology: positive results allowed to classify the effusions as “septic” (and therefore as “not FIP”), independent of their cytological or physicochemical features;histology: histological features consistent with diseases other than FIP allowed to classify the effusion as “non FIP”, independent of the cytological or physicochemical features of the effusions. Conversely, histological findings consistent with FIP [2], especially if associated with positive immunohistochemistry or PCR for FCoVs on tissues, allowed to classify the effusions as “almost confirmed FIP”;PCR: positive FCoV PCR on effusions, performed on the same sample submitted for the cytological and physicochemical analysis with results reported separately from the cytological and physicochemical description and therefore not included in the original report allowed to classify the effusions as “almost confirmed FIP”

Finally, the samples were classified as “non FIP” when other clinical or laboratory results were consistent with diseases other than FIP (e.g., leukemia/lymphoma, hemolytic anemia, pancreatitis, hepatic failure).

### 2.2. Statistical Analysis

Statistical analysis was performed using the Analyse-it software for Microsoft Excel (Analyse-it Software Ltd., Leeds, UK). The comparison of TNCC, LDH activity and LDH/TNCC ratio recorded in the different types of effusions were compared to each other using a non-parametric ANOVA test for independent samples (Kruskall Wallis test), followed by a non-parametric t-test for independent samples (U-Mann Whitney test) to compare paired groups. The same tests were used to compare the results of TNCC, LDH activity and LDH/TNCC ratio recorded in the groups with different likelihood of FIP, either on the basis of data of effusions or on the basis of data of effusions integrated with additional laboratory information of each cat.

In order to assess the discriminating power of TNCC, LDH or LDH/TNCC ratio to support a diagnosis of FIP based on the likelihood of FIP determined in step 2, for each operating point (e.g., each point value recorded in the study population) we calculated the number of true positives, false positives, false negatives and true negatives defined as follows:true positives: samples from cats classified as FIP “probable” or “almost confirmed”, that for each analyte or parameter had values higher than the operating point;false positives: samples from cats classified as “not FIP”, “FIP unlikely” or “FIP possible”, that for each analyte or parameter had values higher than the operating point;false negative: samples from cats classified as FIP “probable” or “almost confirmed”, that for each analyte or parameter had values lower than the operating point;true negatives: samples from cats classified as “not FIP”, “FIP unlikely” or “FIP possible”, that for each analyte or parameter had values lower than the operating point.

For each operating point, Sensitivity (Sens), specificity (Spec), and positive likelihood ratio (LR+) were calculated using standard formulae [32]. Then, a receiver operating characteristic curve (ROC curve) was generated by plotting Sens vs 1-Spec, and the area under the curve (AUC) was calculated. The AUC’s of the three markers (TNCC, LDH and LDH/TNCC ratio) were then compared to each other and the cut-off values with equal sensitivity and specificity as well as those with highest specificity or positive likelihood ratio were then calculated [32]. The ROC curve analysis was then repeated on data generated after the third step of analysis listed above, considering as “positive” the samples belonging to the group “FIP probable/almost confirmed” and as “negative” the samples belonging to the group “FIP unlikely/absent”.

## 3. Results

### 3.1. Caseload

Over a 5-year period, we performed 1949 physicochemical and cytological analyses of feline effusions. We excluded 1268 samples based on the exclusion criteria listed in the material and methods section (Figure 1). Among the 681 remaining samples, 33 were excluded since the total nucleated cell count was not available. Therefore, 648 effusion samples (395 thoracic; 199 abdominal; 9 pericardial; 45 of unknown site of sampling) were included in this study.

### 3.2. Classification of Samples

The distribution of samples according to the criteria used in the three steps of analysis is summarized in Table 3.

Step 1 (distribution of samples based on physicochemical analysis and cytology of the effusions). Most of the samples were classified as “protein-rich transudates” (*n* = 174), “neoplastic” (*n* = 120), or “FIP effusions” (*n* = 117), “chylous effusions” (*n* = 78), and “septic exudates” (*n* = 64). “lymphocyte-rich transudates” (*n* = 32), “non septic exudates” (*n* = 31), “hemorragic effusions” (*n* = 17), and “pure transudates” (*n* = 15) were less represented. 

Step 2 (distribution of samples based on the physicochemical and cytological probability of FIP). Among the types of effusions that, according to the individual presentation, may be consistent with more than one category of probability for FIP, the majority of “chylous effusions” and of “pure transudate” were included in the category “not FIP”, while “lymphocyte-rich transudates” were equally distributed among “not FIP” or “unlikely”. The majority of “protein-rich transudates” were classified as “not FIP” or “unlikely”. However, about half of the cases were included in the categories of “possible FIP” or “probable FIP”. Conversely, most of the “non septic exudates” were considered as “possible” FIP or “probable FIP” and one of these as “almost confirmed FIP” based on positive FCoV PCR. Among the FIP effusions, only a few cases were considered as “possible”, mostly due to the old age of the cats, while the majority belonged to the category of “probable FIP” or “almost confirmed FIP”.

Almost half of the samples included in this study were not consistent with FIP, and the remaining samples were almost equally distributed among “unlikely”, “possible” and “probable” or “almost confirmed”. Ultimately, about 15% of cases (101/648) could be classified as effusions collected from cats with probable FIP, in most cases with the typical cytological pattern. Only in a minority of cases the effusions were classified as “non septic exudate” or “protein rich transudates” and included in the “probable” group because of their cytological appearance. Moreover, in about 17% of these cases (17/101), the presence of FCoV was confirmed by PCR.

Step 3 (distribution of samples based on the overall probability of FIP). As stated in the material and methods section, the overall probability of FIP was based on information from other tests on effusions, blood or tissues, when available. Table 4 summarizes the additional tests available. 

After this further step of analysis, septic, neoplastic, chylous and hemorrhagic fluids, as well as the samples classified as “almost confirmed”, were retained in their classification. The classification of all the other fluids was reviewed according to the results of additional laboratory data (e.g., chemistry, hematology etc.), if available. However, 177 samples were excluded from this further step of analysis, since additional laboratory data were not available or not useful to further modify the probability of FIP. The classification of the samples varied as follows: all the 296 samples classified as “not FIP” remained in the original category, except 2 lymphocyte-rich transudates, that were classified as “unlikely”;among the 37 samples that were originally classified as “unlikely” 10 were downgraded to “not FIP” (one “lymphocyte-rich transudate” and 9 “protein-rich transudates”), and 5 were reclassified as “possible FIP” (one “transudate” and 4 “protein-rich transudates”);among the 63 samples originally classified as “possible FIP” 3 were reclassified as “non FIP” (1 with positive bacteriology and 2 with neoplastic diseases), 27 were downgraded to “unlikely” (19 protein-rich transudates, 7 non septic exudates, 1 FIP effusion), 12 were reclassified as “probable FIP” (7 FIP effusions, 3 protein-rich transudates, 2 non septic exudates ) and 10 with positive PCR for FCoVs, not included in the original report, were then reclassified as “almost confirmed” (7 originally classified as FIP effusions, 2 as protein-rich transudate, 1 as non-septic exudate).among the 58 samples originally classified as “probable FIP” 1 was reclassified as “non FIP” due to a positive bacteriology, 1 was downgraded to “possible FIP” and 18 FIP effusions positive to FCoV PCR were turned to “almost confirmed FIP”;all the 17 effusions that were already classified as “almost confirmed FIP” based on positive FCoV PCR remained in the same class.

According to this new classification, despite the lower number of some type of effusion, the distribution of samples remained similar, and neoplastic samples were the most frequent type of effusions, followed by “FIP effusions”, “chylous effusions”, “protein rich transudates”, and “septic exudate”. "hemorragic effusions”, “non septic exudates”, “lymphocyte transudates” and “pure transudates” were less represented. It is worth noting that most of the fluids that were previously classified as FIP effusions remained in the same category or were FCoV positive at PCR, except in one case that was bacteriologically positive. In addition, some “non-septic exudate” and “protein-rich effusions” that had a positive FCoV PCR were included among the almost confirmed FIP cases. At this step of the analysis, about 2/3 of cases were definitely classified as non FIP, but the proportion of cases classified as “probable” and “almost confirmed” FIP increased to about 20% of the total cases. These were mostly composed by effusions with physicochemical and cytological features consistent with FIP. 

### 3.3. Differences of TNCC, LDH and LDH/TNCC Ratio among Groups

Results regarding the comparison of TNCC, LDH, and LDH/TNCC ratio in the effusions classified according to the cytological and physicochemical appearance (step 1) are reported in Table 5. LDH activity was significantly higher in “septic effusions” than in all the other groups. “Chylous effusions”, “lymphocyte rich transudates”, “protein-rich transudates” and “transudates” had the lowest LDH activity. As regards FIP effusions, LDH activity was significantly lower than that of “septic effusion” and significantly higher than that of all the other groups except “non septic exudates” and "neoplastic effusions”. The TNCCs were significantly different among groups (*p* < 0.001), with the highest values in “septic” and “neoplastic effusions”, that were significantly higher than almost all the other groups; “protein-rich transudates” and “pure transudates” had the lowest cell numbers. The TNCC of FIP effusions was significantly higher than that of “protein-rich transudates” and “transudates”, and significantly lower than that of all the other groups. The highest LDH/TNCC ratio was found in FIP effusions, followed by “non-septic exudates” and “transudates”, while the ratio in the other types of effusions was low to negligible. Interestingly, the LDH/TNCC ratio of FIP effusions was higher than that of all the other groups as shown also in Figure 2. 

Results regarding the comparison of TNCC, LDH, and LDH/TNCC ratio in effusions classified in terms of probability of FIP based on physicochemical and cytological findings (step 2) or also on additional clinical and laboratory data (step 3) are reported in Table 6.

In both steps, the median LDH activity was significantly higher in samples with “probable” or “almost confirmed FIP” (that were not significantly different to each other) than in all the other groups. Among these, the highest LDH values were recorded in the group of cats with no probability of FIP, likely due to the inclusion in this group of “septic” and “neoplastic effusions”, that had high LDH activity. The results of this group were significantly higher than those recorded in the other two groups of effusions. The median TNCC was significantly higher in effusions with no probability of FIP (that included “neoplastic” or “septic effusions”, that, as stated above, had the highest TNCC) than in all the other groups, that were not significantly different to each other. In both cases, the LDH/TNCC ratio progressively and significantly increased with the increased probability of FIP, as shown also in Figure 2. The highest values were recorded in cats with “probable” or “almost confirmed FIP”, whose values were not significantly different to each other.

### 3.4. ROC Curve Analysis for Steps 2 and 3

Figure 3 illustrates the ROC curves obtained for TNCC, LDH activity and LDH/TNCC ratio in steps 2 and 3. In both cases all the ROC curves were significantly different from the line of no discrimination (*p* < 0.001 for all the curves). However, the AUC of the LDH/TNCC ratio (88.7%; 95% CI: 85.5–91.1% for step 2, 87.2%; 84.1–90.3% for step 3) was significantly higher than the AUCs of TNCC (70.5%; 66.5–74.6% for step 2, 77.4%; 73.4–81.4% for step 3) and of LDH activity (69.7%; 65.5–73.9% for step 2, 62.7%; 57.6–67.8% for step 3) (*p* < 0.001 for both the comparisons either in step 2 or in step 3); the AUC of TNCC was significantly higher than that of LDH activity in step 3 but not in step 2.

The diagnostic performances of TNCC, LDH activity and LDH/TNCC ratio are summarized in Table 7. 

For all the tests absolute specificity may be achieved only at very high values for LDH and LDH/TNCC ratio or at very low values for TNCC, and vice versa for the absolute sensitivity. However, the LDH/TNCC ratio had good performances in term of LR+ at moderately high cutoffs and, compared with TNCC and LDH, had the highest Youden index and the best match of sensitivity and specificity at the cut-offs determined by the ROC curves, which were relatively similar in samples classified according to the probability of FIP without (Step 2) or with (Step 3) additional laboratory information. Using this latter approach, however, the cut-off characterized by the highest LR+ was notably lower and had a notably higher LR+, than when the probability of FIP was calculated without information on the results of additional tests.

## 4. Discussion

Although the cytology and the physicochemical features of an effusion are often diagnostic for a given disease or suggestive of a specific physio-pathological process, the inclusion of additional biochemical tests on the same sample may increase the accuracy of the interpretation. For example, analytes such as creatinine, urea, potassium have been historically considered as markers for uroperitoneum, the measurement of lipase or amylase in effusion may support a diagnosis of pancreatitis, the measurement of triglycerides and/or cholesterol is usually employed to differentiate chylous from non chylous effusions, and low glucose concentrations is considered suggestive of bacterial infection [24,25,26,27,28]. More recently, additional specific markers have been recommended by some authors. Among these, the concentration of N-terminal pro-B Natriuretic Peptide (NT-proBNP) may be helpful to differentiate pleural effusions of cardiac origin [33] and the lipoprotein profile has been suggested as a tool to differentiate different types of effusions [34]. In cats with FIP, the albumin to globulin ratio, the percentage of gamma-globulin, and/or an increase of globulin fractions (particularly gamma globulin) have been associated with an increased likelihood of this disease [3,5]. Recently, the measurement of acute phase proteins such as serum amyloid A, haptoglobin and alpha-1-acid glycoprotein in effusions has been recommended as confirmatory tests in suspected FIP [31]. To our knowledge, this is the first study that systematically evaluated the possible clinical utility of LDH and of LDH/TNCC ratios in cats with FIP. 

Our results demonstrate that in effusions classified as FIP based on cytological and physicochemical analysis, the LDH/TNCC ratio is higher than in other types of effusions. This is likely due to the lower cellularity of FIP effusions compared for example with neoplastic effusions, septic effusions and non-septic exudates, that are similarly characterized by a high LDH activity. The TNCC in this study has been obtained using an automated laser counter. This type of counter is known to potentially underestimate the cellularity in FIP effusions, due to the thickness of the fluid and/or to the clumping of cells after contact with some of the reagents used to count the cells [8,35,36]. Higher cell counts can be obtained pre-treating samples with hyaluronidase, that may decrease the viscosity and thickness of the fluid [37]. However, this latter procedure has not be applied to the current samples due to the retrospective nature of this study, and because additional enzymatic treatments of the fluid may theoretically modify other proteins that are present in the effusion, including the enzyme LDH, thus falsely altering the results of biochemical analysis. The TNCC of FIP effusion is known to be lower than other types of inflammatory effusions [24,28], however the cellular count could be affected by the technology used by the cell counter: the result of our study should be therefore revaluated using different hematology analyzers.

The LDH/TNCC ratio was significantly higher in FIP effusions compared with other types of effusions with other etiology, such as “non-specific exudates”, “protein-rich transudates” or “lymphocyte rich transudates”. Since the TNCC of these categories of effusions did not differ from that of FIP effusions, the higher LDH/TNCC ratio does not depend exclusively on the difference in the cellularity but also on the higher LDH activity of FIP effusions. In turn, this may depend on the higher proportion of inflammatory cells (neutrophils and monocytes), that have been shown to be the most important cells producing and releasing LDH in effusions [20,23]. Regardless of the mechanism responsible for this different LDH activity, the highest LDH/TNCC recorded in FIP effusions may have an important diagnostic relevance, since this ratio can be used to differentiate effusions with cytological or physicochemical features that may be borderline between different classification groups. From this perspective, we investigated also the possible differences among samples classified not only based on strict cytological or physicochemical criteria, but also on the subjective interpretation of data recorded during the analysis of the effusions, leading to a scale of probability of FIP. This simulates what usually occurs in diagnostic labs that do not have other clinical or laboratory information about the animals from which samples have been collected. In these cases, clinical pathologists are expected to include in the diagnostic report a conclusive statement about the most likely etiology. The scale of probability based on additional data on the effusions slightly modified the composition of groups, since a few samples classified as “non-septic exudates” or as “protein-rich transudates” were considered as probably affected by FIP and, on the contrary, a moderate proportion of FIP effusions were considered as possibly but not likely affected by FIP. Samples that according to the clinical pathologist were consistent with FIP had the highest LDH/TNCC ratio, suggesting that in practice this ratio may be an optimal additional key to further increase the probability of FIP.

The difference between the LDH/TNCC ratio of “probable”/”almost confirmed” FIP and of other effusions with lower probability of FIP further increased when samples were classified based also on additional laboratory data. In particular, when results of additional tests usually employed to support or exclude a clinical diagnosis of FIP (serum protein electrophoresis, routine hematology, bacteriology of the fluid, histology/immunohistochemistry of masses, PCR for FCoVs on effusions, if not included in the original report) were used to classify the effusions, the large majority of samples cytologically consistent with FIP were ultimately classified as “probable” or “almost confirmed FIP”. This latter group included almost exclusively samples with cytological or physicochemical features consistent with FIP, along with a minority of “non-septic exudates” or “protein-rich transudates”. In all the other effusions the probability of FIP was low to absent.

For an LDH/TNCC ratio higher than 7.54 (the threshold defined by the ROC curve analysis) the probability that the effusion comes from a cat with FIP is 6.59 times higher than the probability that the effusion comes from a cat without FIP. This probability further increases (10.39), at a lower cut-off (2.27) if the sample is classified as “Probable FIP” also on the basis of additional laboratory data. Even at lower cut-offs (0.62 or 0.54), a high LDH/TNCC ratio maximizes the specificity and the sensitivity of the test at value slightly higher than 80%.

This study has some limitations. First, samples have been analyzed during the routine analysis of a commercial laboratory, that receives samples from external clients. Therefore, another possible source of variability of both the TNCC and the LDH activity could be the samples storage over time. However, the express courier of the laboratory, allows the delivery of samples within 24 h. Previous studies demonstrated that the decrease of LDH activity at room temperature for 24 h is minimal [38]; on the other hand, with such a standardization of delivery times, storage artifacts, if any, would have affected all the samples in a similar manner and with similar magnitude. A 24-h storage at room temperature minimally affects the cytological classification of effusions despite some significant decrease of total or differential cell counts. A more difficult identification of bacteria and neoplastic cells, when present, have been reported for stored samples [39]. Therefore, based on this literature information, it is unlikely that storage would have affected the TNCC, the LDH activity, or both. 

Other limitations include the retrospective nature of the study, that does not allow to have additional information on the final diagnosis of the cats, and the subjectivity of the classification of samples in terms of probability by the clinical pathologists. However, both these aspects depend on the nature of the study, that was done on samples processed in a commercial veterinary laboratory, where additional clinical information, sometimes including information on the sampled cavity, that also in the current study was not available in a small proportion of cases, and the information on additional laboratory tests are usually lacking. Necropsy and histological examination, possibly followed by anti-FCoV immunohistochemistry, are still considered the only conclusive tests to confirm FIP [1,2]. However, this information is nowadays difficult to obtain since many vets and owners are used to treat cats with suspected FIP using nucleotide analogues that, although not licensed by regulatory Authorities, are easily available online and widely used. In fact, the good response to these treatments [40,41], prevents obtaining post-mortem tissues from cats with suspected FIP. In the absence of this information the highest level of likelihood of FIP was based on positive PCR on effusions. Ultimately, this test has been reported to have a specificity of 100% [10,12,13,14,15,16,17], and therefore, in cats with positive PCR FIP can be considered as confirmed. However, since the disease was not confirmed histologically, samples with positive PCR in effusions were considered as “almost confirmed”. Therefore, although on one side the approach employed in this study to classify samples based on the probability of disease can be considered a limitation of the study, on the other side it represents what usually happens in routine practice of clinical pathologists working in diagnostic laboratories. Moreover, this approach allowed us to increase the caseload and, in turn, the statistical power of this study. On the contrary, the huge caseload limited the possibility to collect retrospective information on the final diagnosis (e.g., by interviewing each single owner or referring veterinarian on the follow up of sampled cats). Ultimately, a final diagnosis was available only for samples that had cytological features consistent with a given disease (e.g., neoplastic or septic effusions) or that were associated to conclusive diagnostic findings in other specimens (e.g., detection of neoplastic cells in blood or tissues, results of histopathology). In order to bypass the limitations from the lack of additional information, the study was structured in terms of probability of disease, as commonly occurs in routine diagnostic reports. The large sample size of this study minimizes the possible impact of erroneous classification of samples, simulates what commonly occurs in routine practice in diagnostic labs and may be considered as an added value of the study.

## 5. Conclusions

The results of this study confirm our hypothesis that LDH/TNCC ratio is higher in effusions with cytological and physicochemical features consistent with FIP than in other types of effusions. This finding is likely due to the low cellularity of these fluids compared with that of other types of fluids with high LDH activity (septic, neoplastic) and to the higher proportion of neutrophils and macrophages of these fluids compared with other fluids hat may have similar cellularity (“protein-rich transudates or “non-septic exudates”). Regardless of the mechanism responsible for this increase, when the LDH/TNCC ratio of the effusion is high, the clinical pathologist may confidently increase the level of probability of FIP in the diagnostic report of effusions with cytological and chemico-physical features potentially suggestive of FIP, especially if other laboratory or clinical data may be consistent with this diagnosis.

## Figures and Tables

**Figure 1 animals-12-02262-f001:**
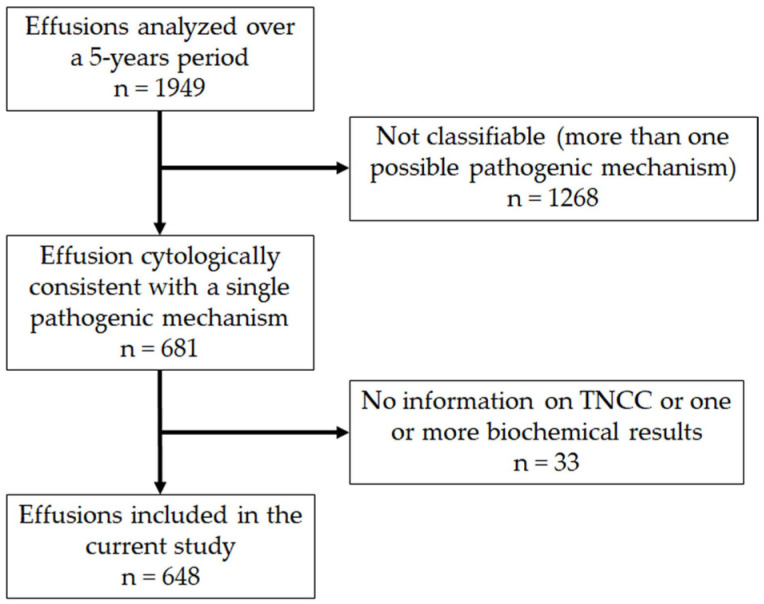
Flowchart summarizing the selection of the caseload according to the exclusion criteria.

**Figure 2 animals-12-02262-f002:**
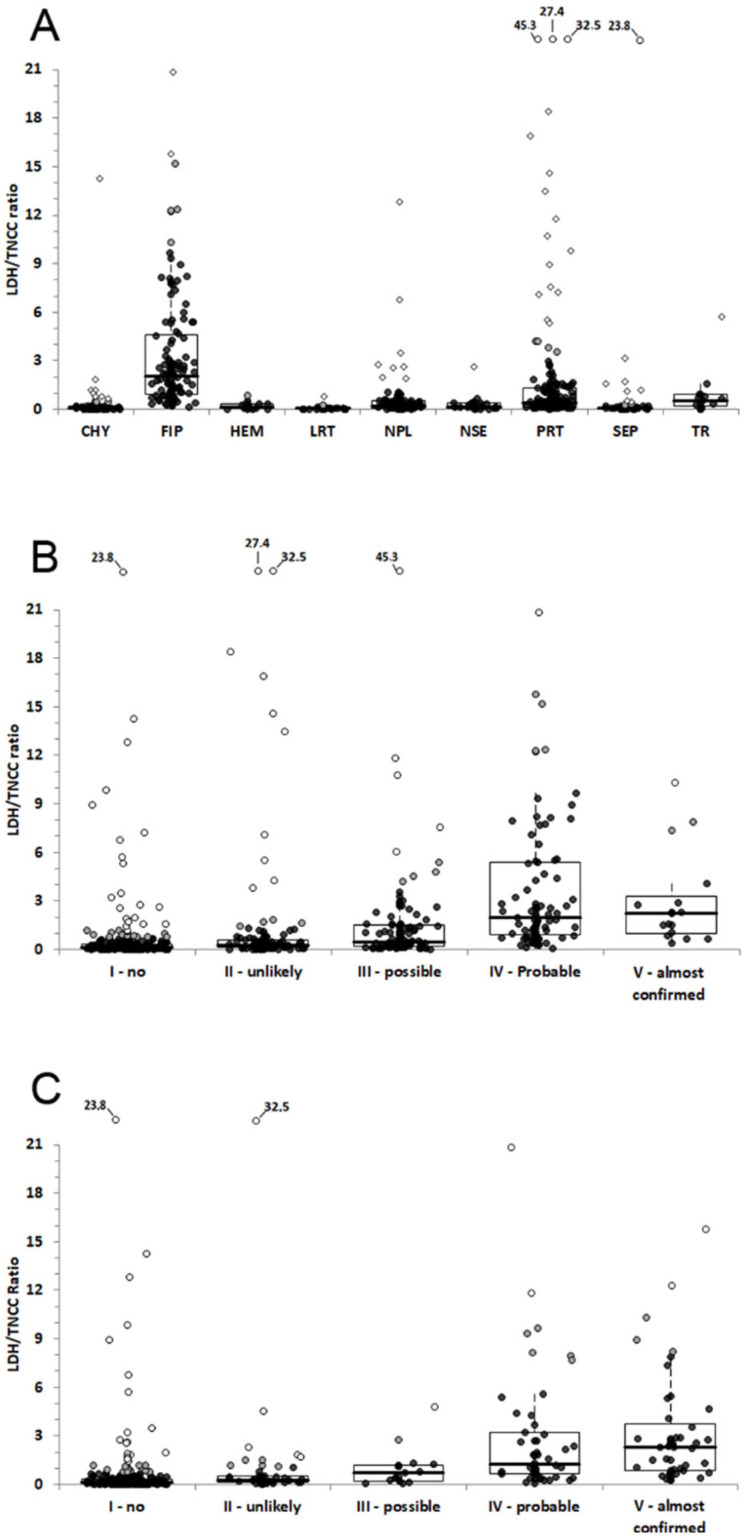
LDH/TNCC ratio recorded in effusions classified based on cytological and physicochemical classification ((**A**), step 1), in terms of probability of FIP based on the subjective interpretation of cytological and physicochemical analysis of the effusions ((**B**), step 2) or on the cytological and physicochemical analysis of the effusions associated with additional laboratory data ((**C**), step 3). The boxes indicate the first to third interquartile range (IQR), the horizontal lines indicate the median value, whiskers extend to further observation within the first quartile minus 1.5 × IQR or to further observation within the third quartile plus 1.5 × IQR. Black dots indicate values not classified as outliers; grey dots indicate the near outliers (values exceeding the third quartile ± (1.5 × IQR)) and white dots the far outliers (values exceeding the third quartile ± (3.0 × IQR). In order to expand the boxes and whiskers, the scale has been limited to a LDH/TNCC ratio of 21, and far outliers that exceeded the limit of the scale have been reported on the top of each graph, with the corresponding point value.

**Figure 3 animals-12-02262-f003:**
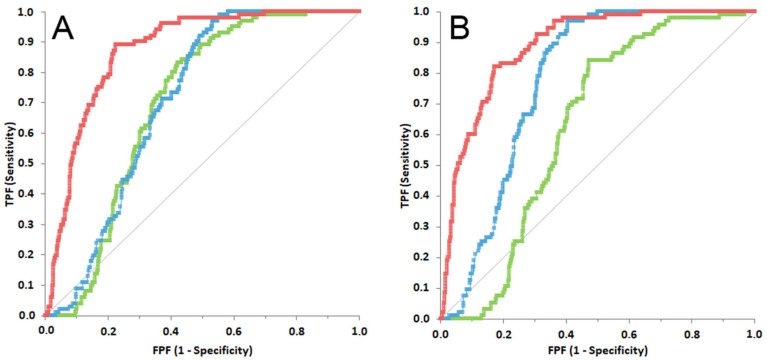
Receiver operating characteristic curve built using the values of LDH/TNCC ratio (in red), LDH (in green) and TNCC (in light blue) calculated based on the results of step 2 ((**A**), probability of FIP based only on cytological and chemico-physical analysis of the effusion) or of step 3 ((**B**), probability of FIP based on cytological and chemico-physical analysis of the effusion and on additional signalment or laboratory data). The central grey line represents the no-discrimination line.

**Table 1 animals-12-02262-t001:** Classification of effusion based on TNCC, chemical analyses and cytology.

Type of Effusion	Description
Pure transudate (protein-poor transudate)	Clear to yellow and transparent; low SG ^1^ (<1.015); Negligible/low protein content (<20 g/L); variably low cellularity (generally <1.5 × 10^9^/L); mixed cell population (mesothelial cells, rare neutrophils, macrophages, and/or lymphocytes)
Protein-rich transudate	Clear to yellow and transparent; SG >1.015); Moderate/high protein content (>20 g/L); variably low cellularity (<5.0 × 10^9^/L); mixed cell populations (mesothelial cells, variable amount of non-degenerated neutrophils, macrophages, and/or lymphocytes)
Lymphocyte-rich transudate	Clear to yellow and transparent; variable SG (1.015–1.030); variable protein content, low triglyceride content with a triglycerides/cholesterol ratio <1; variable cellularity; prevalent population of small lymphocytes without morphological abnormalities, lower amount of non-degenerated neutrophils, macrophages and mesothelial cells
Chylous	White to pink (milky) and turbid; variable SG; variable protein content; high triglyceride content (>1 g/L) or triglycerides/cholesterol ratio >1.0); variable cellularity; prevalent population of small lymphocytes, variable amount of foamy macrophages and neutrophils
Neoplastic	Variable gross appearance, SG and protein content; variably (moderate to high) cellularity; presence of atypical cells of various origin (mostly epithelial or lymphoid)
Hemorrhagic	Bloody appearance; variable SG; high protein content and cellularity; presence of blood cells possibly associated with rare mesothelial cells or macrophages
Non specific exudate	Yellowish, transparent to turbid, possibly with fibrin clots; high SG (>1.015); high protein content (>20 g/L); variable cellularity (>5 × 10^9^/L); prevalent population of neutrophils, mostly non degenerated, in the absence of bacteria, and variable amount of macrophages and lymphocytes
Septic	Yellowish to brownish, usually turbid, possibly with fibrin clots; high SG (>1.015); high protein content (>20 g/L); high cellularity (>5 × 10^9^/L); prevalent population of neutrophils, mostly degenerated, with phagocytosed bacteria, low number of macrophages and lymphocytes, possibly with scant amount of protein granules on the background
FIP	Yellowish, transparent, possibly with fibrin clots; high SG (>1.015); high protein content (>20 g/L) with albumin:globulin ratio <1.0; moderate to low cellularity (generally <5 × 10^9^/L); prevalent population of neutrophils, mostly non degenerated, in the absence of bacteria, and variable amount of macrophages and lymphocytes; presence of proteinaceous granular background

^1^ SG: specific gravity.

**Table 2 animals-12-02262-t002:** Probability of FIP based on results of effusion.

Effusion	Not FIP	Unlikely	Possible	Probable	Almost Confirmed
Hemorragic	All				
Neoplastic	All				
Septic	All				
Chylous	Typical milky appearance and high prevalence of small lymphocytes	Moderate milky appearance, low cellularity and presence of neutrophils along with lymphocytes and monocytes			
Pure transudates	Very low cellularity	Moderate cellularity and predominance of neutrophils			
Lymphocyte-rich transudates	High cellularity and almost absolute presence of lymphocytes	Low cellularity and moderate amount of neutrophils along with lymphocytes			
Protein-rich transudates	Negligible cellularity	Low cellularity and low proportion of neutrophils (lower than mononuclear cells)	High cellularity and moderate proportion of neutrophils (similar to mononuclear cells)	High cellularity, high proportion of neutrophils (higher than mononuclear cells) and proteinaceous background	
Non Septic exudates		Moderate-high cellularity and low proportion of neutrophils compared with macrophages and lymphocytes	Moderate cellularity and prevalence of neutrophils without granular background	Moderate-low cellularity, prevalence of neutrophils and proteinaceous background	Moderate-low cellularity, prevalence of neutrophils and proteinaceous background, positive FCoV PCR
FIP			FIP effusions with low amount of proteinaceous background or from old cats (>3 years)	Typical gross and cytologic appearance but without information on FCoV PCR or with negative FCoV PCR	Typical gross and cytologic appearance and positive FCoV PCR

**Table 3 animals-12-02262-t003:** Classification and distribution of samples.

Step 1: Groups Formed Based on Cytological and Chemico-Physical Analysis of the Effusion		Step 2: Probability of FIP Based Only on Cytological and Chemico-Physical Analysis of the Effusion					Step 3: Probability of FIP Based on Cytological and Chemico-Physical Analysis of the Effusion and on Additional Signalment or Laboratory Data					
Classification	TOTAL	Not FIP	Unl	Pos	Prob	Almost conf	Not FIP	Unl	Pos	Prob	Almost conf	TOTAL
Chylous	78	76	2				76					76
FIP	117			22	79	16	1	1	3	44	42	91
Hemorragic	17	17					17					17
Lymphocyte-rich transudate	32	15	17				5	4				9
Neoplastic	120	120					120					120
Non septic exudate	31		8	19	3	1		8	1	3	1	13
Protein-rich transudate	174	22	79	71	2		15	37	12	3	2	69
Septic	64	64					64					65
Transudate	15	10	5				10	1	1			11
TOTAL	648	324	111	112	84	17	308	51	18	50	45	471

Unl = unlikely; Pos = possible; Prob = probable; conf = confirmed.

**Table 4 animals-12-02262-t004:** Number and distribution of additional tests available.

Classification	H/C	SPE	PCR	Bact	FCoV	Cyto	Histo	Other	No Tests
Chylous	24	20	3	52		5	2	11	16
FIP	46	68	49	56	9	3		9	13
Hemorragic	4	4		9		2		2	4
Lymphocyte-rich transudate	9	9	6	15		4		5	7
Neoplastic	28	32	5	55		18		15	38
Non septic exudate	7	11	14	16	3	1		2	18
Protein-rich transudate	50	54	20	82	5	7	2	13	43
Septic	15	17	2	45	3		1	6	10
Transudate	7	8		9	1			3	1
TOTAL	190	223	99	339	22	40	5	66	140

H/C = hematology and clinical chemistry; SPE = serum protein electrophoresis; PCR = PCR for FCoV genome; Bact = bacteriological analyses; FCoV = anti-FCoV serology by immunofluorescence; Cyto = cytology on masses or other tissues; Histo = histology; Other = additional tests (e.g., serology for FIV or FeLV, Urinalysis, acute phase protein testing, hormones, etc.).

**Table 5 animals-12-02262-t005:** Comparison of TNCC, LDH, and LDH/TNCC ratio in the effusions classified according to the cytological and physicochemical appearance (step 1). Results are expressed as mean ± standard deviation, median and between parenthesis I–III interquartile ranges.

Type of Effusion	LDH (U/L)	TNCC × 10^9^/L	LDH/TNCC Ratio
Chylous (Ch)	692 ± 877^FIP,He,LRT,Npl,NSE,PRT,Se,Tr^ 387 (246–818)	9.2 ± 12.1^FIP,Npl,NSE,PRT,Se,Tr^ 6.2 (2.2–9.8)	0.37 ± 1.63^FIP,Npl,NSE,PRT,Tr^ 0.08 (0.03–0.18)
FIP	3419 ± 3008^Ch,He,LRT,PRT,Se,Tr^ 2211 (1314–4771)	1.6 ± 1.1^Ch,He,LRT,Npl,NSE,PRT,Se,Tr^ 1.3 (0.6–2.3)	3.38 ± 3.70^Ch,He,LRT,Npl,NSE,PRT,Se,Tr^ 2.01 (0.94–4.58)
Hemorragic (He)	2027 ± 3842^Ch,FIP,LRT,Npl,NSE,PRT,Se,Tr^ 908 (357–1724)	10.7 ± 10.7^FIP,PRT,Se,Tr^ 7.3 (4.5–13.3)	0.22 ± 0.22^FIP,LRT,PRT,Tr^ 0.15 (0.06–0.34)
Lymphocyte-rich transudate (LRT)	844 ± 1607^Ch,FIP,He,Npl,NS,Se,Tr^ 236 (131–465)	10.1 ± 11.2^FIP,Npl,NSE,PRT,Se,Tr^ 6.4 (2.6–12.0)	0.10 ± 0.14^FIP,He,Npl,NSE,PRT,Tr^ 0.05 (0.02–0.10)
Neoplastic (Npl)	4628 ± 5502^Ch,He,LRT,PRT,Se,Tr^ 2368 (996–5570)	22.9 ± 27.0^Ch,FIP,LRT,PRT,Se,Tr^ 12.9 (5.0–31.1)	0.56 ± 1.39^Ch,FIP,LRT,PRT,Se^ 0.21 (0.09–0.50)
Non septic exudate (NSE)	4010 ± 6570^Ch,He,LRT,PRT,Se,Tr^ 1673 (1038–3043)	29.8 ± 71.4^Ch.FIP.LRT,PRT,Se,Tr^ 7.9 (6.6–17.8)	0.28 ± 0.47^Ch,FIP,LRT,PRT,Se,Tr^ 0.15 (0.08–0.37)
Protein-rich transudate (PRT)	698 ± 1145^Ch,FIP,He,Npl.NSE,Se,Tr^ 286 (162–670)	1.2 ± 1.1^Ch.FIP.He,LRT,Npl,NSE,Se,Tr^ 0.9 (0.4–1.7)	1.99 ± 5.38^Ch,FIP,He,LRT,Npl,NSE,Se^ 0.42 (0.17–1.30)
Septic (Se)	18861 ± 17,307^Ch,FIP,He,LRT,Npl,NSE,PRT,Tr^ 15,786 (8403–26,868)	217.6 ± 158.7^Ch,FIP,He,LRT,Npl,NSE,PRT,Tr^ 218.7 (57.0–323.9)	0.59 ± 3.00^FIP,Npl,NSE,PRT,Tr^ 0.07 (0.04–0.12)
Transudate (Tr)	366 ± 605^Ch,FIP,He,LRT,Npl,NSE,PRT,Se^ 99 (57–358)	0.7 ± 1.0^Ch,FIP,He,LRT,Npl,NSE,PRT,Se^ 0.3 (0.2–0.8)	0.88 ± 1.40^Ch,FIP,He,LRT,NSE,Se^ 0.50 (0.23–0.89)
	*p* < 0.001	*p* < 0.001	*p* < 0.001

Superscripts indicate the type of effusion that are significantly different.

**Table 6 animals-12-02262-t006:** Comparison of TNCC, LDH, and LDH/TNCC ratio in effusions classified in terms of probability of FIP based on physicochemical and cytological findings (step 2) or based also on additional clinical and laboratory data (step 3). Results are expressed as mean ± standard deviation, median and between parenthesis I–III inter-quartile ranges.

Probability of FIP	Step 2			Step 3		
	LDH (U/L)	TNCC × 10^9^/L	LDH/TNCC ratio	LDH (U/L)	TNCC × 10^9^/L	LDH/TNCC ratio
Not FIP	5772 ± 10768^Un,Pos,Pro^ 1204 (357–6052)	54.8 ± 108.7^Un,Pos,Pro,Conf^ 9.7 (3.3–37.4)	0.57 ± 1.98^Un,Pos,Pro,Conf^ 0.11 (0.06–0.33)	6059 ± 10,968^Un,Pos,Conf^ 1383 (398–7121)	57.4 ± 110.9^Un,Pos,Pro,Conf^ 10.1 (3.7–40.5)	0.56 ± 1.97^Un,Pos,Pro,Conf^ 0.12 (0.06–0.34)
Unlikely	1010 ± 2991^No,Pos,Pro,Conf^ 269 (139–849)	7.5 ± 38.8^No^ 1.2 (0.4–3.0)	1.63 ± 4.93^No,Pro,Conf^ 0.24 (0.10–0.61)	813 ± 1473^No,Pro,Conf^ 319 (182–845)	4.3 ± 8.6^No^ 1.4 (0.4–4.0)	1.17 ± 4.54^No,Pos,Pro,Conf^ 0.24 (0.13–0.50)
Possible	1721 ± 2995^No,Un,Pro,Conf^ 807 (234–1861)	4.2 ± 10.21^No^ 1.2 (0.5–2.6)	1.66 ± 4.58^No,Pro,Conf^ 0.47 (0.19–1.53)	850 ± 1002^No,Pro,Conf^ 415 (223–1208)	1.7 ± 1.9^No^ 0.8 (0.4–2.8)	0.99 ± 1.19^No,Un,Pro,Conf^ 0.69 (0.22–1.21)
Probable	3491 ± 3143^No,Un,Pos^ 2440 (1228–4771)	1.7 ± 1.4^No^ 1.2 (0.6–2.6)	3.62 ± 4.08^No,Un,Pos^ 1.99 (0.92–5.37)	3223 ± 2979^Un,Pos^ 2278 (1189–4432)	2.1 ± 1.7^No^ 1.5 (0.7–2.8)	2.87 ± 3.85^No,Un,Pos^ 1.26 (0.65–3.25)
A. confirmed	3586 ± 2640^Un,Pos^ 2437 (1700–4969)	1.8 ± 1.4^No^ 1.5 (1.0–2.1)	2.96 ± 2.87^No,Un,Pos^ 2.20 (1.00–3.30)	3515 ± 3023^No,Un,Pos^ 2092 (1474–5218)	1.7 ± 1.4^No^ 1.3 (0.8–2.3)	3.20 ± 3.45^No,Un,Pos^ 2.30 (0.87–3.72)
	*p* < 0.001	*p* < 0.001	*p* < 0.001	*p* < 0.001	*p* < 0.001	*p* < 0.001

Superscripts indicate the type of effusion that are significantly different.

**Table 7 animals-12-02262-t007:** Diagnostic performances of TNCC, LDH activity and LDH/TNCC ratio.

Step of Analysis	Parameter	100% Se	100% Sp	Max LR+		Youden Index				Equal Se and Sp		
				Cut-Off	LR+	Index	Cut-Off	% Se	% Sp	Cut-Off	% Se	% Sp
Step 2	TNCC × 10^9^/L	<6.7	<0.1	<2	1.92	0.438	<5.0	97.0	46.8	<1.6	65.3	65.4
	LDH (U/L)	>185	>115,889	>1615	2.04	0.411	>1108	83.2	58	>1679	66.3	66.2
	LDH/TNCC ratio	>0.09	>45.29	>7.54	6.58	0.668	>0.62	89.1	77.7	>0.72	79.2	79.2
Step 3	TNCC × 10^9^/L	<7.9	<0.1	<3.0	2.60	0.564	<5.2	96.8	59.6	<2.5	69.5	70.0
	LDH (U/L)	>91	>115,889	>1108	1.79	0.371	>1108	84.2	52.9	>1815	61.1	60.9
	LDH/TNCC ratio	>0.09	>32.46	>2.27	10.39	0.651	>0.62	82.1	83.0	>0.54	82.1	81.9

Se: sensitivity; Sp: specificity.

## Data Availability

Raw data in excel are available upon request.

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
