# Peer review of "Utility of the Ratio between Lactate Dehydrogenase (LDH) Activity and Total Nucleated Cell Counts in Effusions (LDH/TNCC Ratio) for the Diagnosis of Feline Infectious Peritonitis (FIP)"

_animals, 2022, doi:10.3390/ani12172262_

Round 1

Reviewer 1 Report

The authors propose the ratio LDH/TNCC as a new and complementary approach for the diagnosis of FIP. FIP is a challenging disease concerning its diagnosis and this information may provide a valuable tool. Despite that, there are some limitations related to this study and a few points that are not clear in the article:

-       There are a few typos and incomplete phrases in the introduction and in the discussion;

-       The approach to sort out and select the different cases is not totally clear: it should be clarified if all the analysis were performed at the time of sample arrival to the lab or if, after preservation, some of these were performed specifically for this study;

-       It is also not clear if all samples were tested by RT-PCR or PCR for possible final diagnosis of FIP. The article suggests that the total number of confirmed cases is 17. This is a rather low number of cases and limits the conclusions that can be taken. If possible, a PCR to confirm if the remaining cases are PCR positive or negative would be beneficial as the other effusions can’t totally be confirmed as FIP cases or not; Although I understand the limitation of this study it is not possible to be 100% sure that the comparison that was performed is between positive samples and negative samples for this disease if PCR was not performed in all the cases.

-       It’s also not totally clear how the other effusions were classified (for example, in case of sepsis, were all samples submitted to bacteriology?)

-       Why did the authors consider cases of effusion of unknown origin?

-       Was it possible to check information from each of these cases? Which was the final diagnosis, treatment, survival, post mortem findings in case the animal died?

Author Response

We would like to thank the reviewers for their comments and suggestions, that in the current version have been addressed as listed below (author responses in red)

Reviewer 1

The authors propose the ratio LDH/TNCC as a new and complementary approach for the diagnosis of FIP. FIP is a challenging disease concerning its diagnosis and this information may provide a valuable tool. Despite that, there are some limitations related to this study and a few points that are not clear in the article:

-       There are a few typos and incomplete phrases in the introduction and in the discussion;

Typos and English language have been corrected throughout the text and the manuscript has been revised by an English speaking specialist. We also made a minor change on table 7, where the symbols “<” and “>” were inserted to make easier to understand how the cut-offs must be used in practice

-       The approach to sort out and select the different cases is not totally clear: it should be clarified if all the analysis were performed at the time of sample arrival to the lab or if, after preservation, some of these were performed specifically for this study;

All the samples have been processed just after arrival to the laboratory. This has now been specified in the manuscript (Lines 137-139). Being this a retrospective study, no samples or tests were specifically performed for this study.

-       It is also not clear if all samples were tested by RT-PCR or PCR for possible final diagnosis of FIP. The article suggests that the total number of confirmed cases is 17. This is a rather low number of cases and limits the conclusions that can be taken. If possible, a PCR to confirm if the remaining cases are PCR positive or negative would be beneficial as the other effusions can’t totally be confirmed as FIP cases or not; Although I understand the limitation of this study it is not possible to be 100% sure that the comparison that was performed is between positive samples and negative samples for this disease if PCR was not performed in all the cases.

PCR for FIP was performed in 99 cases (in 30 cases the results were included in the report of cytological and physicochemical analysis, in the remaining 69 cases the results were included in a separate report since the sample went separately to the molecular diagnostic unit). In 17 out of these cases the positive result of PCR was included in the original cytological report and therefore were classified as “confirmed” FIP (now “almost confirmed”, based on the recommendation of reviewer 2) in step 2; in 28 cases, the information about positive PCR was not available when the probability of FIP was estimated only on the basis of the cytological and physicochemical report and therefore these samples were classified as “confirmed” (now “almost confirmed”) in step 3. The number of samples receiving additional tests has now been tabled (new table 4)

-       It’s also not totally clear how the other effusions were classified (for example, in case of sepsis, were all samples submitted to bacteriology?)

Additional bacteriological tests were performed in 339 cases, either to confirm the presence of bacteria seen cytologically or to investigate/exclude the possible presence of bacteria. As for PCR (see the comment above) the number of samples receiving additional bacteriological tests has now been included in the new table 4

-       Why did the authors consider cases of effusion of unknown origin?

We assume that this comment refers to the samples that were submitted to the laboratory without any information on the sampling site (pericardial, pleural, abdominal). As stated in the manuscript, this is a retrospective study based on the routine caseload of a commercial laboratory, where information on the patient or on sample are sometime incomplete. Therefore, also the lack of this information further simulate what routinely happens in these labs and further reinforces the practical power of the current study. A statement stressing that the lack of information on the sampled cavity may occur in routine activity of diagnostic labs has been added to the discussion (lines 723-725)

-       Was it possible to check information from each of these cases? Which was the final diagnosis, treatment, survival, post mortem findings in case the animal died?

No, it was not possible to check clinical information on the samples submitted to the laboratory. This was already stressed in the previous version of the manuscript and it has now better clarified (lines 750-756)

Reviewer 2 Report

Feline infectious peritonitis is a fatal coronavirus infection of cat species. Although therapeutic agents for FIP are becoming commercially available, veterinarians have yet to control this disease. One reason for this is the lack of a rapid diagnostic method for FIP. The authors attempted to establish new diagnostic criteria for FIP by the ratio of lactate dehydrogenase activity (LDH) and total nucleated cell counts (TNCC) in exudates collected from wet type FIP. Although the diagnostic criteria for FIP in this paper are undeniably vague, we believe that the results of this study will be useful in actual clinical practice. Therefore, I strongly recommend publication of this study in animals.

minor points;

line 88: Sentences seem to be broken. Please add or correct the words.

Table 2: "Confirmed" diagnostic criteria for the presence or absence of the gene for FCoV. As the authors know, the confirmed diagnosis of FIP is the detection of FCoV antigen in lesional tissue or cells in the exudate. Since it is not academically correct to describe "confirmed" only by genetic diagnosis, it is recommended to change the term to "almost confirmed". If you have any reason for a confirmed diagnosis based on genetic testing alone for this, please describe it in the discussion. In my experience, FCoV gene has been detected in ascites fluid several times in cases other than FIP (tumor, pyothorax, etc.).

As mentioned in the authors' paper, the highly viscous FIP leachate seems to nonspecifically reduce TNCC. In particular, the storage conditions of the samples should have a significant impact on their numbers. However, I believe that LDH/TNCC can be considered as one of the criteria for FIP diagnosis.

Author Response

We would like to thank the reviewers for their comments and suggestions, that in the current version have been addressed as listed below (author responses in red)

Reviewer 2

Feline infectious peritonitis is a fatal coronavirus infection of cat species. Although therapeutic agents for FIP are becoming commercially available, veterinarians have yet to control this disease. One reason for this is the lack of a rapid diagnostic method for FIP. The authors attempted to establish new diagnostic criteria for FIP by the ratio of lactate dehydrogenase activity (LDH) and total nucleated cell counts (TNCC) in exudates collected from wet type FIP. Although the diagnostic criteria for FIP in this paper are undeniably vague, we believe that the results of this study will be useful in actual clinical practice. Therefore, I strongly recommend publication of this study in animals.

We would thank the reviewer for this kind comment

minor points;

line 88: Sentences seem to be broken. Please add or correct the words.

The sentence has been completed and corrected

Table 2: "Confirmed" diagnostic criteria for the presence or absence of the gene for FCoV. As the authors know, the confirmed diagnosis of FIP is the detection of FCoV antigen in lesional tissue or cells in the exudate. Since it is not academically correct to describe "confirmed" only by genetic diagnosis, it is recommended to change the term to "almost confirmed". If you have any reason for a confirmed diagnosis based on genetic testing alone for this, please describe it in the discussion. In my experience, FCoV gene has been detected in ascites fluid several times in cases other than FIP (tumor, pyothorax, etc.).

We would thank the reviewer for this recommendation. “Confirmed” has been replaced by “almost confirmed” throughout the study, in the tables (that have been reformatted to not split the titles in two rows, when needed) and in the figure, and the rationale for employing the term “almost confirmed” instead of “confirmed” has been added to the discussion (lines 742-745)

As mentioned in the authors' paper, the highly viscous FIP leachate seems to nonspecifically reduce TNCC. In particular, the storage conditions of the samples should have a significant impact on their numbers. However, I believe that LDH/TNCC can be considered as one of the criteria for FIP diagnosis.

The authors appreciate this further comment of this reviewer

Round 2

Reviewer 1 Report

All comments were considered in this new version.